# Byte-level Tokenizers Unavoidably Enable LLMs to Generate Ill-formed UTF-8

**Preston Firestone, Shubham Ugare, Gagandeep Singh, Sasa Misailovic**
University of Illinois Urbana-Champaign
{`pf8`, `sugare2`, `ggnds`, `misailo`}@illinois.edu

## Abstract

Subword tokenization segments input text according to a pre-defined vocabulary to feed it into a language model; the language model, in turn, generates a sequence made from this same vocabulary. The members of the vocabulary can be built of code points or bytes. Using code points means that all members of the vocabulary are valid UTF-8 characters. However, it also requires thousands of initial members to achieve acceptable coverage of inputs and more than a million to entirely avoid out-of-vocabulary errors. Beginning with bytes, on the contrary, avoids out-of-vocabulary errors with only 256 initial members of the vocabulary, but the members of the vocabulary and sequences of them are not guaranteed to be valid UTF-8. Sequences that are not valid UTF-8 break code that assumes its input to be valid UTF-8. Applications of language models that operate under this assumption must account for the breakage thereby introduced.

In this paper, we formalize tokenization using monoid theory and prove that byte-level tokenizers with vocabularies smaller than the full Unicode space inevitably face either out-of-vocabulary issues or generate invalid UTF-8 sequences. We demonstrate formally that attempting to incrementally convert tokens back to a string and interpret the results as UTF-8 gives different results than converting the whole sequence of tokens at once. This formal result causes real-world bugs; in some cases we discover bugs whose existence was predicted by the theoretical result. We evaluate mitigations for the problem identified and provide case studies of major foundation models, serving engines, and constrained generation systems.

## 1 Introduction

Systems for processing natural language usually begin by cutting their input, a long series of symbols, into snippets. Traditionally, cutting has been done along boundaries from prior knowledge of the language, such as words, morphemes, or phonemes. In the last ten years, natural language processing has moved to using *"subword" tokenization*, where the cutting is derived directly from the corpus rather than programmed manually (Mielke et al., 2021). These approaches begin with a maximally (minimally) cut training corpus and remove (add) cuts until some stopping condition is reached. The set of snippets in the resultant cutting of the training corpus contains the vocabulary that will be used to segment future inputs.

When beginning this process, one generally has two choices of granularity: character-based, as introduced by Sennrich et al. (2016); Kudo & Richardson (2018), or byte-based, as introduced by Radford et al. (2019); Wang et al. (2020). The character-based approach begins with a vocabulary made up of characters and cuts inputs into snippets of characters. The byte-based approach cuts the input into snippets of bytes. The character-based approach always generates valid characters, but will encounter characters that are out of its vocabulary unless the vocabulary includes all Unicode *code points*, each representing a numerical value that maps to a specific Unicode character.

Detokenization should be *a morphism*: combining two token sequences and detokenizing them should yield the same result as detokenizing each sequence separately and combining the results. This property is crucial for reliable constrained generation and token-by-token streaming interfaces, where incremental processing must align with final outputs. We show that byte-level detokenizers are not in general morphisms because of the post-processing that converts the bytes into

Table 1: Unicode code points and bytes representing the first word of the Rigveda, अग्निमीळे, transliterated *agnim iḷe* and meaning "I praise Agni [fire]". The top row contains the Unicode code points making up the sequence. Note that several of the characters are broken down into seperate base and combining code points (Unicode Technical Commitee, 2025, §2.11). The bottom row is the UTF-8 bytes encoding the top row. The squiggles ≀ indicate the tokenization of this word according to `cl100k_base`, GPT-4's tokenizer's vocabulary.

| अ | ग | ् | न | ि | म | ी | ळ | े |
|---|---|---|---|---|---|---|---|---|
| ≀E0A4≀85≀ | ≀E0A4≀97≀ | ≀E0A58D | E0A4≀A8≀ | ≀E0A4BF | E0A4≀AE≀ | ≀E0A580≀ | ≀E0A4≀B3≀ | ≀E0A587≀ |

characters for manipulation by further processes. Table 1 shows the tokenization of a string by a byte-level tokenizer and illustrates the problem. If we were detokenizing incrementally, the first token would be the bytes 'E0 A4', then '85', then 'E0 A4', and so on. Any token that begins or ends in the middle of a character encoding, which in the bottom row of Example 1 is any token whose beginning or end is not aligned with the vertical lines separating characters, cannot be decoded as valid UTF-8, and attempting to do so will lead to an error. Then detokenizing the two tokens E0 A4 and 85 separately will give two errors, but detokenizing the two tokens together will give the character अ.

**Contributions:** The main contributions of our paper are:

★ We introduce a new formalism for tokenization, based on cutting sequences into snippets based on a vocabulary, based on the segmentations of Berglund & van der Merwe (2023) and the monoids and stochastic maps of Gastaldi et al. (2024).

★ Using our new formalism, we prove an impossibility result limiting vocabularies made up of byte-level tokens: if not all UTF-8 encoded characters are present in the vocabulary, it will either be able to generate sequences that are ill-formed UTF-8 or encounter UTF-8 encoding forms that are not in its vocabulary.

★ We show that decoding a sequence of tokens as UTF-8 can violate formal assumptions about the properties of tokenizers, such as their being lossless or homomorphisms.

★ We categorize families of foundation language models according to whether they use a character-level or byte-level tokenizer, study the impact that our theoretical results have had on serving engines and constrained generation systems, and find and fix bugs in existing tools.

## 2 Background

Tokenization is a preprocessing step of almost all language models (LMs). Most existing tokenizers work by breaking an input text into discrete segments of a vocabulary. Deciding what the members of the vocabulary should be and determining which of the many possible segmentations to return are major distinctions among existing tokenization algorithms (see Schmidt et al., 2024).

### 2.1 Representing formal languages by monoids

Following Gastaldi et al. (2024), Sakarovitch (2009) and Lothaire (1997), we treat languages as monoids (algebraic structures). We develop the formalism here in sketch to suit our needs and refer the reader to these references for detail.

**Definition 1** (Monoid). A monoid is a triple composed of a set of elements, a binary operation on members of that set that is associative but not commutative, and an identity element such that applying the binary operation to the identity element and any other member of the monoid results in that member. Monoids' binary operation is associative for all of its members:

$$(s_1 \cdot_\Sigma s_2) \cdot_\Sigma s_3 = s_1 \cdot_\Sigma (s_2 \cdot_\Sigma s_3) \tag{1}$$

The identity element results in no change when paired with any member of the monoid.

$$s \cdot_\Sigma \epsilon_\Sigma = \epsilon_\Sigma \cdot_\Sigma s = s \tag{2}$$

Set theoretical operations ($\subset, \subseteq, =, \ldots$) are defined between monoids by comparing their sets.

We refer to monoids by Greek capital letters, their members by subscripted lowercase Latin letters, the binary operation is a subscripted dot, and the empty string is a subscripted epsilon. A sequence of members of a monoid is referred to by a lowercase Greek letter. Subscripts and binary operations are left out when context makes the meaning clear. We refer to the binary operation as concatenation, and when multiple members of $\Sigma$ have been concatenated, as in Equation (1), we refer to the result as a sequence.

**Definition 2** (Free monoid). The **free monoid** generated by a set $\Sigma$, denoted $\Sigma^*$, is the set of all finite sequences (strings) of elements from $\Sigma$, including the empty string (when $n = 0$):

$$\Sigma^* = \{s_1 s_2 \dots s_n \mid n \geq 0, s_i \in \Sigma \text{ for all } i\} \tag{3}$$

## 2.2 Relationship between finite set and free monoid

Here we establish a relationship between a vocabulary and the set of all the sequences it can generate. The vocabulary is of finite size ($|\Sigma| \in \mathbb{N}$) and the set of all sequences is the free monoid over that vocabulary ($\Sigma^*$). In particular, we need to be able to compare the free monoids of two potentially disjoint vocabularies in order to make claims about what sequences can and can't be generated by those vocabularies. We state here two lemmas and corollary that will be useful in the next sections.

**Lemma 1.** *If all the members of a vocabulary $\Sigma$ are in the vocabulary $\Delta$, then all the sequences that can be made of members of $\Sigma$ can be made of members of $\Delta$:*

$$\Sigma \subseteq \Delta \rightarrow \Sigma^* \subseteq \Delta^*.$$

**Corollary 1.** *If there is some free monoid $\Sigma^*$ that contains a member not in some other free monoid $\Delta^*$, then the monoid $\Sigma$ also contains a member not in $\Delta$:*

$$\Sigma^* \not\subseteq \Delta^* \rightarrow \Sigma \not\subseteq \Delta.$$

The converse of this corollary can be true or false. A counter example is where $\Sigma$ contains all of $\Delta$ and some members of $\Delta^*$: clearly $\Sigma^*$ is equal to $\Delta^*$, so the implication cannot hold in all cases. We can recuperate the converse of Corollary 1 in a weak form as Lemma 2.

**Lemma 2.** *If some monoid $\Sigma$ contains a member not in some free monoid $\Delta^*$, the the free monoid $\Sigma^*$ of the first monoid $\Sigma$ also contains a members not in the free monoid $\Delta^*$:*

$$\Sigma \not\subseteq \Delta^* \rightarrow \Sigma^* \not\subseteq \Delta^*.$$

## 2.3 Mappings and homomorphism

We shall map sequences back and forth between different monoids in order to describe tokenization and detokenization of inputs.

**Definition 3** (Stochastic map). A **stochastic map** $\kappa$ from a monoid $\Sigma$ to a monoid $\Delta$ is a function from $\Sigma$ to the set of probability distributions on $\Delta$ (Gastaldi et al., 2024).

We do not work explicitly with probability distributions in this paper, so our notation will act as if maps between monoids are functions. *We do not introduce the assumption that a mapping always returns the same value.* The deterministic case can be represented in a stochastic framework by assigning a probability of one to the expected tokenization and zero to all others, so we retain the generality of not including this assumption.

Finally we introduce the concept of a morphism.

**Definition 4** (Morphism). A map $\kappa$ from a monoid $\Sigma$ to $\Delta$ is a **morphism** if and only if, for all $\sigma, \sigma'$ in $\Sigma$,

$$\kappa(\sigma \cdot_\Sigma \sigma') = \kappa(\sigma) \cdot_\Delta \kappa(\sigma') \qquad \text{and} \qquad \kappa(\epsilon_\sigma) = \epsilon_\delta. \tag{4}$$

# 3 Tokenization and Monoids: UTF-8 Breaks Homomorphism

In Subsection 3.1, we introduce a novel formalization of tokenization in terms of cutting the input into snippets found in a pre-established vocabulary, building to Theorem 1, showing that tokenization is only a morphism in one direction. In Subsection 3.2, we provide an impossibility result,

Proposition 1, showing that a byte-level vocabulary, unless it contains at least all the encoding forms of UTF-8, will either encounter inputs that are not in its vocabulary or be able to generate sequences that are not well-formed UTF-8. Finally, in Subsection 3.3, we show that decoding the bytes a model generates as UTF-8 is not a morphism.

## 3.1 Binding and cutting monoids

**Definition 5** (Bound monoid). A **bound monoid** of a given monoid is a finite subset of that monoid and is represented by a superscripted circle.

$$(\Sigma^\circ \subset \Sigma) \wedge (|\Sigma^\circ| \leq n, n \in \mathbb{N}) \tag{5}$$

We also need a way of keeping track of the smaller sequences out of which larger sequences are constructed by introducing separated markers between.

**Definition 6** (Cut monoid). The **cut monoid** of a given monoid is that monoid, to whose set of members the new symbol $\wr$ has been added, and whose members have been prepended and postpended by it. It is indicated by a superscripted squiggle

$$\Sigma^\wr = \{\wr\sigma\wr \,|\, \sigma \in \Sigma\} \tag{6}$$

When discussing a member of a cut monoid, we refer to each of the subsequences between the $\wr$ signs as **snippets** of the longer sequence. We collapse adjacent squiggles for legibility: $\wr\wr \rightarrow \wr$.

Our formalism of cutting is most similar to the treatment of Berglund & van der Merwe (2023), though unlike them we describe the cutting process in terms of monoids and do not refer to particular tokenization algorithms. When applying the freeing operation (see Definition 2), the bounding operation, and the cutting operation to monoids built from the same original "alphabet", the binary operation and identity element remain the same by the natural insertion of all these monoids into their common basis. This will be necessary for our definition of a tokenizer.

**Definition 7** (Tokenizer). We call a tokenizer any pair of mappings $(\tau, \kappa)$ between some monoids $\Sigma^*$ and $\Sigma^{*\wr\circ*}$ such that:

$$\tau : \Sigma^* \rightarrow \Sigma^{*\wr\circ*} \qquad \kappa : \Sigma^{*\wr\circ*} \rightarrow \Sigma^* \qquad \forall \sigma \in \Sigma^* \quad \kappa\tau\sigma = \sigma. \tag{7}$$

In the semantics introduced in this section, $\tau$ cuts its argument such that each snippet of the cut sequence is a member of some finite set of snippets defined in advance, the vocabulary $\Sigma^{*\wr\circ}$. Each given tokenizer $\tau$ is parameterized by the vocabulary $\Sigma^{*\wr\circ}$ it is defined over, but we omit this detail in the notation because it is not relevant to our needs. $\tau$ is a stochastic map in the sense of Definition 3: it returns a probability distribution over $\Sigma^{*\wr\circ*}$ (Gastaldi et al., 2024; Ahia et al., 2023). $\kappa$ joins the cut sequence by removing all the cut operators.[1] The definition of tokenization and homomorphisms imply the following important theorem:

**Theorem 1.** *$\kappa$ is a morphism, but $\tau$ is not.*

*Proof. Joining is homomorphic:* The result of concatenating two cut sequences and removing the cut marks is always identical to removing the cut marks and concatenating them: the cut symbols are removed and the resultant string is identical to its uncut version.

*Cutting is not homomorphic:* Take some $s_0 \cdots s_n \in \Sigma^*$ and cut it into some $\wr s_0 \cdots s_{m-1} \wr s_m \cdot s_{m+1} \wr s_{m+2} \cdots s_n \wr$. It is impossible to cut $s_0 \cdots s_m$ and $s_{m+1} \cdots s_n$ such that there is not a cut after $s_m$ and before $s_{m+1}$, so the concatenation of any separate cutting of the two sequences must include the subsequence $s_m \wr s_{m+1}$, even though this was not necessary when cutting the entire sequence $s_0 \cdots s_n$. $\qquad\square$

We shall make extensive use of this theorem in what follows.

---

[1]Mnemonically, $\tau$ stands for τέμνω, to cut, and $\kappa$ stands for κολλάω, to glue.

## 3.2 The UTF-8 Encoding Scheme

Humans interact with text in the form of characters, but computers interact with numbers. To that end, the Unicode Standard maps each graphical character of each human writing system to a unique natural number (Unicode Technical Commitee, 2025). In the standard, each cell depicts a character ("Abstract character" in Unicode jargon) and an associated number ("Code point" in Unicode jargon.)

**Definition 8** (Code point). The Unicode code points are the values in the range 0 through 1,114,111; we refer to the monoid of the code points as $\Gamma = \{\gamma | \gamma \in [0, 1114111]\}$ (Unicode Technical Commitee, 2025, D9).

Each code point is associated with a character with an appearance, properties, and so on. To store them in a computer, we must encode the members of $\Gamma$ as members of B, stored as a sequence of bytes in memory. UTF-8 is the dominant encoding scheme (Unicode Technical Commitee, 2025).

**Definition 9** (UTF-8). The UTF-8 encoding forms are a finite number sequences $\beta$ of $B^*$, each one to four bytes long. We refer to the monoid of UTF-8 forms as Y, which is some $B^{*\circ}$. We say that $|Y| = |\Gamma|$, so there exists a one-to-one (or bijective) mapping between $\Gamma$ and Y, allowing us to put each member of $\Gamma$ into the machine as a unique member of Y. The sequences of $B^*$ that are not in $Y^*$ are ill-formed; the members of $B^*$ that are in $Y^*$ are well-formed.[2]

**Proposition 1.** *Any vocabulary $B^{*l\circ}$ that contains a member $\beta$ that is ill-formed UTF-8 ($\sigma \notin Y^*$) will be able to generate ill-formed sequences:*

$$B^{*l\circ} \nsubseteq Y^* \rightarrow B^{*l\circ*} \nsubseteq Y^*. \tag{8a}$$

*Any vocabulary that does not contain at least all well-formed UTF-8 character encodings will not be able to generate all well-formed UTF-8 sequences:*

$$Y \nsubseteq B^{*l\circ*} \rightarrow Y^* \nsubseteq B^{*l\circ*}. \tag{8b}$$

*Proof.* Lemma 2 says that if a monoid ($B^{*l\circ}$ in Equation (8a) and Y in Equation (8b)) that contains members not in the free monoid of some other monoid ($Y^*$ in Equation (8a) and $B^{*l\circ*}$ in Equation (8b)), then the free monoid of that monoid ($B^{*l\circ*}$ in Equation (8a) and $Y^*$ in Equation (8b)) contains members not in the free monoid of the other monoid. We simply substitute $B^{*l\circ}$ and Y for the $\Sigma$ and $\Delta$ of Lemma 2. □

Proposition 1 establishes that byte-level tokenizers face an unavoidable issue: they must either include invalid UTF-8 sequences in their vocabulary or be unable to represent all possible Unicode text. This fundamental limitation creates a trade-off between vocabulary size, coverage, and UTF-8 validity that affects all byte-level tokenizers.

## 3.3 Enforcing UTF-8 breaks homomorphism

**Definition 10** (Decoding). The process of cutting a sequence $\beta$ in $B^*$ into some $\beta^l$ in $B^{*l}$ such that the number of snippets of $\beta^l$ that are members of Y is maximized is called **decoding**.[3] UTF-8 guarantees that there is exactly one way of **decoding** each $\beta$ in $B^*$. **Decoding** can be understood tokenizing an input $\beta$ using some deterministic $\tau$ whose vocabulary is $Y^l$.

**Definition 11** (Encoding). The opposite direction, going from a sequence of $Y^*$ to a sequence of $B^*$, is called **encoding**.[4] This can be understood as detokenizing using some $\kappa$ on a member of $Y^{l*}$ to produce the corresponding member of $B^*$ by $Y^{l*}$'s natural insertion into $B^*$ (recall, $Y^{l*}$ is syntactic sugar for a particular $B^{*\circ l*}$).

---

[2]See Appendix B for more detail.

[3]The direction from $B^*$ to $Y^*$ is "decoding" because our ultimate goal is to recover the abstract characters in $\Gamma$ encoded by the members of Y that we recover from $\beta$.

[4]The direction from $Y^*$ to $B^*$ is called "encoding" because we produce the bytes $\beta$ that encode the sequence of $\Gamma^*$ represented by this particular sequence of $Y^*$.

---

**Algorithm 1** Buffer bytes that are not yet valid encoding forms until they are completed by subsequent tokens; emit characters as and when they can be successfully decoded.

---

**Require:** *buffer* ← a queue for tokens.
 1: **function** INCREMENTALDETOKENIZATION(*token*)
 2:    ENQUEUE(*buffer, token*)
 3:    **if** ISWELLFORMEDUTF8(*buffer*) **then**
 4:      *decoded* ← DECODEUTF8(*buffer*)
 5:      CLEAR(*buffer*)
 6:      **return** *decoded*
 7:    **else**
 8:      **return** $\epsilon$

---

Our definition of encoding and decoding differs from that of the Unicode Standard, because we are eliding, on purpose, the difference between $\Gamma$ and $\Upsilon$. Because of a one-to-one map between $\Gamma$ and $\Upsilon$, we can safely ignore the difference between them: converting from one to the other is trivial.

**Theorem 2.** *UTF-8 decoding is not a morphism, but UTF-8 encoding is.*

*Proof.* Theorem 2 follows from Theorem 1 and Definition 10. Decoding is understood as tokenizing a sequence of bytes using a vocabulary made up of all and only the valid UTF-8 forms (the members of $\Upsilon$). The cutting function $\tau$ is known not to be homomorphic. UTF-8 encoding, on the other hand, is simply the natural insertion of the members of $\Upsilon$ into a byte sequence in $B^*$ (Unicode Technical Commitee, 2025, §3.10, D95) □

## 4 Sealing the Leaks in UTF-8 Decoding

Theorem 2 says that decoding UTF-8 is not a morphism. Since the final step of byte-level detokenizers is to decode the concatenated tokens as if they were UTF-8, and since UTF-8 decoding is not morphism, this causes detokenization as a whole to not be a morphism. We saw an example of detokenizing that was not a morphism in Section 1. In Subsection 4.1 we present Algorithm 1, which patches over this problem, and in Subsection 4.2 we study cases of foundation models, serving engines, and constrained generation systems implementing Algorithm 1 and affected by Theorem 2

### 4.1 General solution to incrementally detokenizing byte-level vocabularies encoded using UTF-8.

In order to avoid the breakage caused by erroneously attempting to decode ill-formed bytes, we introduce Algorithm 1, which incrementally detokenizes tokens (proceeding in one direction, always appending tokens). When a token or tokens are ill-formed UTF-8, it caches them until subsequent tokens arrive that make the previous ones well-formed. Algorithm 1 can be understood as a transducer in the sense of Cognetta & Okazaki (2024), where the input alphabet is sequences of bytes and the output alphabet is sequences of code points.

Algorithm 1 is not a full solution, but it restores enough functionality to solve major issues, discussed in Subsection 4.2. The time cost of Algorithm 1 is trivial compared the time it takes for the neural network to generate a token and for that token to be copied from the GPU to the CPU for further processing. However, as shown in Appendix B, it is easy to create sequences of bytes that never contain any well-formed subsequences; the memory cost of Algorithm 1 is therefore unlimited (see Section 6).

Table 2 provides an example. Algorithm 1 consumes the tokens in the bottom row one at a time from left to right, emitting characters as and when it can. The caching behavior is shown beginning at the token ⸗EA⸗, after which the Algorithm consumes two more tokens, ⸗99⸗ and ⸗AE⸗, before emitting the character ⊛ that the three bytes taken together encode. With out Algorithm 1, one would not be able to generate ⊛, the multiocular O.

Table 2: The result of applying Algorithm 1 to the Old Church Slavonic word много⃰читїй, transliterated *mnogoočitii* and meaning "many-eyed". The top row is the characters in Γ, and the bottom row the sequence of Υ* that encodes them. The squiggles ⸨ indicate the tokenization of `cl100k_base`.

| м | ного | | | ⃰ | чит | | ї | й |
|---|------|---|---|---|-----|---|---|---|
| ⸨D0BC⸩ | ⸨D0BDD0BED0B3D0BE⸩ | ⸨EA⸩ | ⸨99⸩ | ⸨AE⸩ | ⸨D187D0B8D182⸩ | ⸨D1⸩ | ⸨97⸩ | ⸨D0B9⸩ |

## 4.2 Case studies

**Tokenization strategies of foundation models**  We begin by categorizing existing models according to whether their tokenizers are character-level or byte-level; that is, whether their vocabulary is some $Υ^{*⸨○}$ or some $B^{*⸨○}$. Papers do not consistently report the tokenization style they use: it very often is passed over; generally, series of models (e.g. Llamas, Gemmas...) reuse the same tokenization style of their predecessor. Ablations are seldom performed [5], and usually only a single vocabulary is prepared. The information in Table 3, when not described in the paper introducing the model, has been determined from publicly available code or by reference to earlier papers in the series. All the tokenizers we studied use byte-pair encoding (Gage, 1994) to train their vocabularies, suggesting the potential for experimentation with varying approaches. It would be too expensive to train each of these series of models, which already exist in several sizes and variants, using multiple different tokenization strategies; the power of BPE has attained the status of a folk theorem and is taken for granted.

A middle way exists between byte-level and character-level tokenization, introduced by SentencePiece (Kudo, 2024, v0.1.9). This is referred to as a byte-fallback approach, and the release notes for version 0.1.9 cite Wang et al. (2020), though the byte-fallback approach is distinct from that used by Wang et al. (2020): rather than beginning with a vocabulary of 256 bytes, the SentencePiece byte-fallback concept begins with a set of code points and learns the vocabulary from them as usual. During inference, when the trained tokenizer encounters a stretch of bytes for which it has no token, it replaces those bytes with strings that represent their value (e.g. "<0x89>" for $(89_{16})$ and so on). This retains the property of character-level vocabularies that they only generate well-formed UTF-8 and lets the model recover some information about the unknown characters. This approach is not exempt from Theorem 2: consider that a tokenizer without ï in its vocabulary would tokenize that character as "<0xD1><0x97>". This violates Equation 7, which says that tokenizing and detokenizing an input should return exactly that input. Tokenizers that do not satisfy this property have their own problems that are outside of the scope of this paper.

Table 3: Foundation models and the tokenizers they use.

| Model | Tokenizer Type |
|-------|----------------|
| OpenAI since GPT-2 (Radford et al., 2019; Brown et al., 2020; OpenAI et al., 2024) | Byte-level Byte-Pair Encoding |
| Qwen, Qwen2.5, Qwen3 (Bai et al., 2023; Yang et al., 2025b;a) | Byte-level Byte-Pair Encoding [6] |
| Llama 1, 2 (Touvron et al., 2023a;b) | Character-level with Byte Fallback [7] [8] |
| Llama 3 (Grattafiori et al., 2024) | Byte-level Byte-Pair Encoding |
| Mistral, Mixtral (Jiang et al., 2023; 2024) | Character-level with Byte Fallback [9] |
| Gemma 1, 2, 3 (Mesnard et al., 2024; Team et al., 2024; 2025) | Character-level with Byte Fallback [10] |
| OLMO, OLMo 2 (Groeneveld et al., 2024; OLMo et al., 2025) | Byte-level Byte-Pair Encoding [11] |
| Phi-4 (Microsoft et al., 2025) | Byte-level Byte-Pair Encoding |

**Existing constrained generation systems and non-homomorphic tokenizers**  Constrained generation techniques (Scholak et al., 2021; Poesia et al., 2022; Willard & Louf, 2023; Ugare et al., 2024; 2025; Banerjee et al., 2025; Loula et al., 2025) are used to restrict language model outputs to adhere to specified rules.

We examined various grammar-constrained generation systems and tested them to discover their behavior during partial generation decoding. Among popular grammar-constrained generation

---

[5]With the notable exception of OLMo2 (OLMo et al., 2025), which ablates on the size of the tokenizer's vocabulary, but not on byte- versus character-level tokenization.

tools, we found that Synchromesh Poesia et al. (2022) and SynCode Ugare et al. (2024) encountered issues when grammars included non-ASCII characters such as emojis or mathematical symbols such as '∀'. Both tools use character-based parsers rather than byte-based ones, which created this vulnerability. We reported these issues to the SynCode authors and worked with them to fix these problems in accordance to the Algorithm 1.

To evaluate SynCode's ability to handle non-ASCII Unicode characters, we conducted an experiment using emoji generation task. We selected a subset of the TweetEval emoji dataset Barbieri et al. (2020), filtering for three common emoji classes. The task required the model to generate exactly one emoji character in response to a given tweet, adhering to a constrained grammar specification. This evaluation directly tested SynCode's handling of multi-byte UTF-8 sequences, which was a limitation in earlier versions. The prompt template instructed the model to analyze tweets and respond with exactly one emoji from the allowed set (Listing 2 in Appendix D). We evaluated this task across 100 examples from the TweetEval test set on two versions of SynCode: v0.2.0 which used a character-level finite state machine (FSM), and the current version which implements our proposed fix in accordance to Algorithm 1.

Table 4 presents the results of our evaluation. SynCode v0.2.0 with its character-level FSM failed on all 100 examples, resulting in a 100% crash rate and 0% accuracy. In comparison, the current implementation after the fix processed all examples without crashes, achieving 62% accuracy in emoji prediction.

Table 4: Performance comparison between SynCode versions on emoji generation task

| Metric | SynCode v0.2.0 | Current Version |
|---|---|---|
| Accuracy | 0% | 62% |
| Crash Rate | 100% | 0% |

**Serving Engines** Algorithm 1 is a way to correctly incrementally detokenize and is the standard approach serving engines use to do so. A version of Algorithm 1 first appeared in basaran hyperonym (2023). It was introduced to Huggingface TGI by a user's issue (Hugging Face, 2025; 0x1997, 2023). It percolated thence to vLLM (Kwon et al., 2023; Yard1, 2023), OpenLLM (bentoml, 2025; jeffwang0516, 2023), and SGLang (Zheng et al., 2024; hnyls2002, 2024). The code deployed to these engines was produced ad hoc to solve the issue presented without any analysis of its impacts or of the broader implications brought out in Section 3. Our Algorithm 1 is a simplification of the procedures used in these engines, and we provide a theoretical framework for motivating the algorithm and examining its deficiencies.

## 5 Related Work

**UTF-8 Challenges in Tokenization.** Rahman et al. (2024, §II.D) discuss the challenges of UTF-8, both because it encodes characters as sequences of varying length, and many characters are encoded by more than one byte. The latter challenge cannot be avoided unless one wants to limit one's character space to 256 unique characters, but the former issue is unique to UTF-8 and distinguishes it from other encoding schemes, UTF-16 and UTF-32, which encode all code points in two and four bytes respectively. (Petrov et al., 2023; Ahia et al., 2023) show that byte-level and character-level tokenization introduce severe discrepancies among languages, due both to the varying lengths of bytes or code points used to represent text in various languages, and because of the number of tokens popular tokenizers need to tokenize input text.

**Tokenization Notation and Terminology.** The process we call "cutting" is a standard step in tokenization but is represented variously in the literature. Sennrich et al. (2016) use (|) or a space; Gastaldi et al. (2024) use (⌄); Schmidt et al. (2024) and Bostrom & Durrett (2020) use a space; Koo et al. (2024) use (|); Kudo (2018) uses (/); Kudo & Richardson (2018) wrap tokens with square brackets; Cognetta & Okazaki (2024) use ␣; andGeng et al. (2024) use color to distinguish tokens. We follow Berglund & van der Merwe (2023) in using (≀), though we extend their notation by requiring that cut sequences begin and end with (≀).

We do not require that $\tau$ always return the same cutting for a given input: *pace* Gastaldi et al. (2024), we do not assume that our tokenizer is deterministic. This assumption is not necessary to

the results of this paper, and leaving it out makes them apply to the stochastic tokenizers Gastaldi et al. (2024) describe, of which deterministic tokenizers are merely a special case.

**Morphisms in Tokenization.** The term homomorphism is used by Geng et al. (2024) for what we call a morphism; they leave out the requirement that the empty string map to the empty string. For Sakarovitch (2009), a homomorphism is a bijective morphism, which Theorem 1 shows tokenizers are not. Gastaldi et al. (2024) call **multiplicative** what we refer to as a morphism; with the following additional constraint that it map non-empty sequences to non-empty sequences, it also has a *trivial kernel*: $\delta \neq \epsilon_\Delta \rightarrow \kappa(\delta) \neq \epsilon_\Sigma$. Lothaire (1997) and Sakarovitch (2009) call a mapping that respects Equation (4) a **morphism** and include the additional requirement that it map the identity member to the identity member, $\kappa(\epsilon_\Delta) = \epsilon_\Sigma$.

**Properties of Tokenizers.** The property of Equation 7 is optional in some tokenizer definitions. Gastaldi et al. (2024) call such a tokenizer **exact**. Rajaraman et al. (2024) call such a mapping **consistent** and only concern themselves with such tokenizers. For Kudo & Richardson (2018) such a tokenizer is **lossless**. We define our tokenizer as linked to a single vocabulary and say that it returns a probability distribution over the tokenizations that are possible using that vocabulary. In terms of Gastaldi et al. (2024), such a tokenizer is a stochastic map such that the probability of any tokenization that contains a token that is not in the tokenizer's vocabulary is zero.

**Subword Tokenization Methods.** A cutting tokenizer like ours is a generalization of a subword tokenizer (Sennrich et al., 2016; Kudo, 2018; Kudo & Richardson, 2018). The performance of subword tokenizers is contested. The reader is directed *inter alia/* to (Gallé, 2019; Zouhar et al., 2023) for a favorable impression and to (Bostrom & Durrett, 2020; Schmidt et al., 2024; Chai et al., 2024a) for a negative one. The best representatives of the negative camp are tokenizer-free models that work either directly on input characters (e.g. Tay et al., 2022; Clark et al., 2022) or bytes (e.g. Xue et al., 2022; Pagnoni et al., 2024), or on images of input text (e.g. Salesky et al., 2021; Rust et al., 2023; Chai et al., 2024b). Limisiewicz et al. (2024); Hofmann et al. (2022) examine adapting subword tokenization to respect morphological boundaries in language to improve performance in non-European languages. Though we must ask why or whether cutting the input in necessary, we do not discuss it here and direct the reader to the copious literature on the subject (e.g. Gastaldi, 2021; Mielke et al., 2021; Rajaraman et al., 2024). We show the prevalence of cutting tokenizers in Section 4.2 and consider this to justify our focusing only on this type of tokenizer.

**Byte-level and Character-level Approaches.** Byte-pair encoding has been studied extensively: Bostrom & Durrett (2020) argue that Unigram is superior to BPE. Gallé (2019) argue that BPE performs highly because it compresses the input, but Schmidt et al. (2024) perform experiments that suggest that compression is not necessary or sufficient for performance in a tokenizer. Zouhar et al. (2023) propose an information theoretic standard, Rényi entropy, for why certain tokenizers perform better than others; Cognetta et al. (2024) supply counter examples to the argument of Zouhar et al. (2023). Libovický et al. (2022) examine the efficacy of tokenizationless character-level machine translation compared to character-level BPE and find that the former performs only at best as well as the latter.

## 6 Conclusions and Future Work

The primary theme of this paper: the law of leaky abstractions (Spolsky, 2002; Kiczales et al., 1992; Kiczales, 1991) is ineluctable, and attempts to overcome it end in tragedy (Kott, 1964). Rather than burying the issue, practitioners should accustom themselves to not expecting the inputs or outputs of their system to be valid UTF-8. Implementers of systems and applications for language models should test their implementations on non-ASCII characters and ensure that they behave properly when an input or generated sequence is ill-formed UTF-8. Authors should be more precise when specifying the tokenization scheme their model uses when describing its architecture. This can be communicated concisely using the terminology advanced in this paper.

Ultimately the goal of the abstractions we present in this paper is to facilitate the untangling of the chains of bytes that encode text in our computers. All humans have the right to interact with computers in their own language and to be able to use computers to manipulate their language,

with equal regard to all languages. This paper makes a small step toward that goal by addressing a leak in one of the abstractions in LLMs.

## Ethics Statement

Davis & Suignard (2014) describes security issues that face systems that interact with Unicode as closely as language model applications and infrastructure must. They describe non-visual exploits such as buffer overflows during encoding or decoding, text comparison, ill-formed input bytes, including several exploits that are particular to UTF-8. Davis & Suignard (2014)'s visual exploits are based on visual spoofs, visually confusable strings: these are two or more different sequences of code points that appear the same to the user (see Unicode Technical Commitee (2025, §3.11) and Davis et al. (2024)). These would be tokenized differently by all tokenizers, because they are not the same byte or code point sequence and so cannot be represented by the same tokens.

Visual spoofs be used for attacks similar to those recently studied by Geh et al. (2025), where models generated radically different responses to varying tokenizations of the same prompt; in some cases Geh et al. (2025) broke safety and alignment restrictions trained into models. A visual spoof would circumvent a mitigation Geh et al. (2025) suggest for their attack: providers of language models as a service could prohibit users from tokenizing their own texts and require them to submit well-formed UTF-8 in order to avoid adversarial tokenizations. Visual spoofs could produce the same effect of adversarial tokenization without the user needing to have direct control over the tokenization process.

When discussing how to decode UTF-8 sequences, the Unicode standard offers the following forboding words: "[s]ilently ignoring ill-formed sequences is strongly discouraged because joining text from before and after the ill-formed sequence can cause the resulting text to take a new meaning. This result would be especially dangerous in the context of textual formats that carry embedded program code" (Unicode Technical Commitee, 2025, C10).

As discussed in Subsection 4.1, Algorithm 1 is vulnerable to memory leaks caused by pathologically ill-formed inputs. Any byte-level vocabulary will be able to produce these types of inputs, as shown in Appendix B. In future work we shall experimentally attack the systems we studied in Subsection 4.2.

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

# A Mapping bytes to unicode and back

Huggingface distributes tokenizer vocabularies as UTF-8-encoded files. [12] Since programmers tend to work with UTF-8-encoded strings rather than directly with byte sequences, it is convenient to be able to represent arbitrary byte sequences, such as those produced by arbitrarily clumping bytes, as UTF-8 sequences. [13]

The solution to handling non-UTF-8 tokens with software that expects UTF-8 the community has adopted since GPT-2 (Radford et al., 2019) is to map each byte value onto a Unicode Code Point. Most of the Code Points between $00_{16}$ and $FF_{16}$ are printable characters, so though the result will be an ugly mix of semantically insignificant characters, the result will be made up entirely of printable characters. The transition between the middle and bottom rows of Example 5 exemplifies these transformations. The problem is that not all of the Code Points between 0 and FF are printable: many are control or whitespace characters. As these are encountered they are replaced by the next available Code Point starting at U+0100. Luckily the entire block U+0100–U+17F0 is printable characters. See the attached file for the relevant code blocks from Unicode Technical Commitee (2025). 📌

Tokenizers built using a converted version of the initial vocabulary of bytes are not generally homomorphic due to this pre- and post-processing. The components of the Huggingface transformers and tokenizer libraries implement this behavior through most of their models. In the Python (slow) implementations of tokenizers for Huggingace models, the function implementing this behaviod appears in 24 files (Wolf et al., 2020). In the Rust (fast) implementation in the tokenizers library, the enbyting and debyting process is implemented in the ByteLevel pretokenizer and decoder (Huggingface, 2025, `pre_tokenizers/byte_level.rs#L14`). The call to the Rust standard library's String function `from_UTF8_lossy` (The Rust Foundation, 2025) when enbyting debyted tokens(Huggingface, 2025, `pre_tokenizers/byte_level.rs#L174`) is the source of the U+FFFD � REPLACEMENT CHARACTER in detokenization and the ultimate breaker of the homorphism in Huggingface's tokenizers.

Table 5: The first two words of *The Building of Skadar* (Serbian: Зидање Скадра), "Град градила", transliterated *"Grad gradila"* ("The city was built"). The top row is the sequence of code points representing this word and the bottom row is the sequence of bytes that are the UTF-8 encoding of this sequence of code points. In the terms of Section 3, the top row is a member of $\Gamma^*$, the middle row is a member of $Y^*$, and the bottom row is the debyted version of the middle row. The squiggles ⁊ in the bottom two rows indicate the cutting that turn a member of $Y^*$ into a member of $B^{*⁊\circ*}$. The $B^{*⁊\circ}$ in question is Qwen2.5's vocabulary.

| Г | р | а | д | | г | р | а | д | и | л | а |
|---|---|---|---|---|---|---|---|---|---|---|---|
| ⁊D093⁊ | ⁊D180 | D0B0 | D0B4⁊ | ⁊20 | D0B3⁊ | ⁊D180 | D0B0 | D0B4⁊ | ⁊D0B8⁊ | ⁊D0BB | D0B0⁊ |
| ⁊Ðĵ⁊ | ⁊ÑĢ | Ð° | Ð´⁊ | ⁊Ġ | Ð³⁊ | ⁊ÑĢ | Ð° | Ð´⁊ | ⁊Ð⁊ | ⁊Ð» | Ð°⁊ |

Comment from the paper:

> Despite its name, reference [Byte Pair Encoding] implementations often operate on Unicode code points and not byte sequences. These implementations would require including the full space of Unicode symbols in order to model all Unicode strings. This would result in a base vocabulary of over 130,000 before any multi-symbol tokens are added. This is prohibitively large compared to the 32,000 to 64,000 token vocabularies often used with BPE. In contrast, a byte-level version of BPE only requires a base vocabulary of size 256. (Radford et al., 2019, § 2.2)

Docstring of the function.

---

[12] In HTTP-speak: `content-type: text/plain; charset=UTF-8`

[13] A notable exception is OpenAI, who after GPT-2 have worked directly with byte sequences that are not guaranteed to be valid UTF-8.

**Algorithm 2** Code from GPT-2. Defines an injective but not surjective mapping between the set of natural numbers from 0 to 255 and the set of natural numbers. The numbers $21_{16}, \ldots 7E_{16}, A1_{16}, \ldots AC_{16}, AE_{16}, \ldots, FF_{16}$ are mapped to themselves. The remaining numbers are mapped in order to $100_{16}$ through $143_{16}$.

```python
def bytes_to_unicode():
    bs = (
        list(range(ord("!"), ord("~") + 1))
        + list(range(ord("¡"), ord("¬") + 1))
        + list(range(ord("®"), ord("ÿ") + 1))
    )
    cs = bs[:]
    n = 0
    for b in range(2**8):
        if b not in bs:
            bs.append(b)
            cs.append(2**8 + n)
            n += 1
    cs = [chr(n) for n in cs]
    return dict(zip(bs, cs))
```

> Returns list of UTF-8 byte and a corresponding list of unicode strings. The reversible bpe codes work on unicode strings. This means you need a large # of unicode characters in your vocab if you want to avoid UNKs. When you're at something like a 10B token dataset you end up needing around 5K for decent coverage. This is a signficant percentage of your normal, say, 32K bpe vocab. To avoid that, we want lookup tables between UTF-8 bytes and unicode strings. And avoids mapping to whitespace/control characters the bpe code barfs on. (OpanAI, 2019, src/encoder.py)

The API of the Huggingface library provides direct access to the underlying representations of their tokens, represented as code points via Algorithm 2, through the methods `convert_ids_to_tokens` and `convert_tokens_to_ids`: these map between indices into the vocabulary and the tokens those indices represent.

An alternative solution to the problem identified as Theorem 2 is not to require that the bytes that go into and come out of the language model be well-formed UTF-8 sequences. Since models interact directly with indices into their lookup table of token embeddings, and these tokens already map on to arbitrary bytes, no change to the model would be necessary to work directly with byte sequences: the code that dealt with the strings going in to and coming out of the model would have to change, but the model itself would be undisturbed. This makes it difficult for human users to interpret what the model has generated as a part of a human language.

## B    Details about UTF-8

The interested reader is directed to Unicode Technical Commitee (2025, §3.9.3) for details of the scheme and Pike & Thompson (1993) for an early account of the encoding scheme. The relevant Wikipedia page `https://en.wikipedia.org/wiki/UTF-8` is also excellent. As of 2025, UTF-8 is used by 98.5% of all websites (W3Techs, 2025). Tables 6 and 7, from the Unicode standard, show how to encode code points as bits and bytes of UTF-8.

Our use of decode and encode is influenced by the Python standard library (https://docs.python.org/3/library/stdtypes.html#str.encode, https://docs.python.org/3/library/stdtypes.html#bytes.decode), which uses the terms as we do. We draw the usage of encode from the Unicode standard (cf. Unicode Code Spec sec. 3.9), since UTF-8 is referred to as an "character encoding scheme" and individual units of it are "character encoding forms". In the Standard an encoded character also refers to any character assigned to a code point (Unicode Core Spec Definition 11, https://www.unicode.org/versions/Unicode16.0.0/core-

Table 6: UTF-8 bit distribution, showing how to convert code points, represented as binary numbers, into the bytes of UTF-8. (Unicode Technical Commitee, 2025, Table 3-6).

| Scalar Value | First Byte | Second Byte | Third Byte | Fourth Byte |
|---|---|---|---|---|
| 00000000 0xxxxxxx | 0xxxxxxx | | | |
| 00000yyy yyxxxxxx | 110yyyyy | 10xxxxxx | | |
| zzzzyyyy yyxxxxxx | 1110zzzz | 10yyyyyy | 10xxxxxx | |
| 000uuuuu zzzzyyyy yyxxxxxx | 11110uuu | 10uuzzzz | 10yyyyyy | 10xxxxxx |

spec/chapter-3/#G22752. See also https://www.unicode.org/reports/tr17/). Our use of the terms is therefore idiosyncratic but, we hope, self-consistent.

Our use of the term decode throughout the paper is intentional and should not be confused with detokenization: decoding refers to the process of interpreting a sequence of bytes as UTF-8 and detokenization refers to the process of turning a sequence of tokens into a sequence of bytes. Almost all occurrences of the term "decode" we find to be consistent with our definition of the term (Definition 10). The only ambiguous use that we find is "we examined various grammar-constrained generation systems and tested them to discover their behavior during partial generation decoding" from section 4.2, where both senses (decoding and detokenizing) could be intended. We will repair this in future revision.

The difficulty is illustrated by the example in Table 1. The Devanagari script used to write Sanskrit (and Hindi, Marathi, Pali, and many more) is an abugida, meaning that consonant letters and vowel letters combine to form consonant-vowel units. Further, the script also allows for diacritical markings on these units. Comparing the text अगनर्मिीळे with the separated abstract characters (Unicode Code Spec Definition 7, https://www.unicode.org/versions/Unicode16.0.0/core-spec/chapter-3/#G2230) in the table will help the reader develop an intuition for the terminological complexity involved in Unicode. It is preferable to adhere to Unicode terminology as closely as possible (which we have not done without error, as we point out in our response to your question 3), because that is the most exact way to discuss the problem. On the other hand, this makes it difficult to introduce the problem at a high level before clarifying matters with definitions. We shall continue to polish this in future revisions and thank you for the critique.

The decision of how to handle the ill-formed parts of a byte sequence when decoding it from UTF-8 lies, usually, with the library programmer and not with the application programmer. To be compliant, "a UTF-8 conversion process is required to never consume well-formed subsequences as part of its error handling for ill-formed subsequences," meaning that the decoding process must recover all well-formed sequences in the input. As long as the well-formed parts of the sequence are left unchanged, a decoding process "is not otherwise constrained in how it deals with any ill-formed subsequence itself" (Unicode Technical Commitee, 2025, §3.9.5). When encountering ill-formed byte sequences, a proccess has three main options: fail entirely, replace the ill-formed code unit with a marker such as �[14], or silently drop the ill-formed sequences.[15] The ill-formed sequences are those cuttings of the $B^{*\circ*\wr}$ that are not in Y. This section is concerned with the impact these problems have in practice.

Several properties of UTF-8 referenced in the main text are immediately obvious by visual inspection of Tables 6 and 7. For example, any sequence made up exclusively of bytes starting 10, that is, in the range [80, BF], can never be well-formed UTF-8. Similarly, no sequence made up exclusively of bytes beginning with 110, 1110, or 11110 can ever contain any well-formed encoded forms. Also, the bytes C0-C1 and F5-FF never appear in UTF-8 at all, so they can also be used to break up well-formed encoding forms.

By convention, code points are represented as hexadecimal numbers preceded by "U+". Any given sequence of $B^*$ either is or isn't in $Y^*$. Any subsequence of a single UTF-8 encoded character that is not the entire entire encoded character is ill-formed. For any given sequence of $B^*$ there

---

[14]See Unicode Technical Commitee (2025, §3.9.6) and van Kesteren (2024, §4.1) for the standard algorithm for this replacement.

[15]Inserting � is known as "replacement" and failure is known as "fatal" in the terms of van Kesteren (2024, §4.1).

Table 7: Well-formed UTF-8 byte sequences and the range of code points they encode (Unicode Technical Commitee, 2025, Table 3-7).

| Code Points | First Byte | Second Byte | Third Byte | Fourth Byte |
| --- | --- | --- | --- | --- |
| U+0000..U+007F | 00..7F | | | |
| U+0080..U+07FF | C2..DF | 80..BF | | |
| U+0800..U+0FFF | E0 | A0..BF | 80..BF | |
| U+1000..U+CFFF | E1..EC | 80..BF | 80..BF | |
| U+D000..U+D7FF | ED | 80..9F | 80..BF | |
| U+E000..U+FFFF | EE..EF | 80..BF | 80..BF | |
| U+10000..U+3FFFF | F0 | 90..BF | 80..BF | 80..BF |
| U+40000..U+FFFFF | F1..F3 | 80..BF | 80..BF | 80..BF |
| U+100000..U+10FFFF | F4 | 80..8F | 80..BF | 80..BF |

exists a unique way to recover the most possible members of Y. The scheme defining Y gives it the characteristic that not all members of B are in it: in fact, only half of bytes (those beginning with 0) are valid UTF-8 sequences on their own. Eleven bytes (C0, C1, F5–FF) are not used at all and so never appear in Y, and the remaining 117 only appear in sequences longer than one byte. The sequences of Y range in length from one to four bytes, and not all sequences of one to four bytes are in Y. By Corollary 1, there are sequences that can be generated by a vocabulary of bytes that are not valid UTF-8.

## C  Proofs

### C.1  Lemma 1

*If all the members of a vocabulary $\Sigma$ are in the vocabulary $\Delta$, then all the sequences that can be made of members of $\Sigma$ can be made of members of $\Delta$:*

$$\Sigma \subseteq \Delta \rightarrow \Sigma^* \subseteq \Delta^*.$$

*Proof.* Assume the opposite to derive a contradiction:

$$\exists \Sigma, \Delta \quad \Sigma \subseteq \Delta \wedge \Sigma^* \nsubseteq \Delta^*.$$

It follows that

$$\exists \sigma \in \Sigma^* \quad \sigma \notin \Delta^*.$$

$\Delta^*$ contains all sequences made up of members of $\Delta$, so any sequence not in it must not be made up of members of $\Delta$. Then

$$\exists s \in \sigma \quad s \notin \Delta$$

Since $\sigma$ is in $\Sigma^*$, all of its factors must be in $\Sigma$. There is a factor of $\sigma$ not in $\Delta$, so there must be a member of $\Sigma$ not in $\Delta$. We assumed that $\Sigma$ was a subset of $\Delta$ and derived a contradiction, which is what we wanted to show. $\square$

### C.2  Lemma 2

*Proof.* By definition, $\Sigma \nsubseteq \Delta^*$ implies that there exists some $\sigma$ in $\Sigma$ that is not in $\Delta^*$. All members of $\Sigma$ are also members of $\Sigma^*$, so the $\sigma$ not in $\Delta^*$ is in $\Sigma^*$. $\square$

## D  Unicode Character Handling SynCode Evaluation Prompt and Grammar

**Listing 1** Grammar specification for emoji generation task using Unicode escape sequences. The three emojis are "a face with heart-shaped eyes", "a face with tears of joy", and "a winking face", respectively.

```
// Lark grammar to validate single emoji output
start: emoji

// Define the 3 emojis from the TweetEval emoji dataset
emoji: "\U0001F60D" | "\U0001F602" | "\U0001F609"
```

**Listing 2** Prompt template for emoji generation task (emoji symbols represented as placeholders)

```
You are evaluating tweets to assign the most appropriate emoji.

INSTRUCTIONS:
1. Read the tweet below carefully.
2. Select the SINGLE most appropriate emoji that captures the
↪   sentiment.
3. Respond with ONLY that emoji - no words or other characters.

The emoji must be one of the 3 valid options from this set:
[heart-eyes] [laughing] [winking]

Tweet: "{tweet_text}"
Your response:
```

