# OpenReview forum: "Byte-level Tokenizers Unavoidably Enable LLMs to Generate Ill-formed UTF-8"
_ICML.cc/2025/Workshop/TokShop — TokShop_

### Official Review · Reviewer_ogDX · 2025-06-06
**Review for Submission 8**

**Rating:** 1
**Confidence:** 4

**Review:**

This paper uses the COLM template, which is specifically not allowed, per the call for papers:

>Authors may not modify the style files or use templates designed for other conferences.

Furthermore, the paper was submitted in a format without line numbers (not the submission version).

I believe this could lead to desk rejection, but I will provide a review. This paper tackles a very important paper for multilingual NLP and tokenization. However, the paper is difficult to follow and does not make a significant contribution above and beyond existing implementations.

The authors do not sufficiently explain what a monoid is, which makes the paper inaccessible to those that aren’t already familiar. The paper also contains too much formal proof content, which is insufficiently explained and motivated. This also contributes to making the paper inaccessible to a broader audience. Furthermore, as the authors acknowledge in the final paragraph of Section 4, the main part of the proposed algorithm is already implemented in major frameworks like vLLM.

The paper should engage with related work identifying ‘ill-formed sequences’, but with different terminology: ‘partial UTF-8 sequences’ (https://aclanthology.org/2024.emnlp-main.649/)  and ‘incomplete tokens’ (https://arxiv.org/pdf/2410.23684).

I have one question about this point:
>the memory cost of Algorithm 1 is therefore unlimited (see Section 6)

Does this potentially lead to issues with implementing this algorithm in practice?

I gave this paper a score of 1 due to the formatting issue. If the program chairs ignore this issue, I recommend a score of 4.

---

### Official Review · Reviewer_MpXm · 2025-06-07
**Provides a theoretical proof of byte level tokenizers generating ill formed UTF-8 which could inform better design of tokenizers / de-tokenizers in future LLMs**

**Rating:** 7
**Confidence:** 4

**Review:**

## Summary:
The paper provides a nice monoid theory based proof of how byte level tokenizers will always run into cases where they generate ill formed UTF-8 sequences which can cause errors in systems built on top of LLMs especially if the system does not account for such error patterns. The authors nicely represent the tokenization and de-tokenization problem into a formal mathematical construct using monoid theory and provide easy to understand explanations of their lemmas and theorems. Finally, they also propose a small algorithmic improvement during decoding to help counter this effect for existing tokenizers and show the benefits of this approach via a quick quantitative experiment.

## Strengths:

- Raises an important issue which is core to the design of current and future LLMs that will be the base of many softwares. This should provoke future LLM architects to incorporate the learnings from this paper.

- I appreciate the authors in how they converted the process of algorithmic tokenization and de-tokenization from the dataset also in the form of a mathematical construct to justify the claim made in the paper in a theoretical manner. Proofs and explanation are sound and easy to understand.

- The performance improvement in Table 4 is really impressive and should be noted.

- The discussion on implications of such errors in the ethics statement was also relevant and would encourage software engineers to build robustness in their systems to avoid such attacks.

## Weaknesses:
- From the discussion on "Servning Engines" on Page 8, its not clear whether all the quoted usages of Algorithm 1 were inspired from the work presented in the paper or were those independently discovered and applied. If its the latter, I am not sure if its suitable to call out Algorithm 1 as a novel contribution in this paper.

### Nits:

- Page 2, Section 2.1: Should be "algebraic" if i understand correctly.

---

### Decision · Program_Chairs · 2025-06-10

Accept